# Music and dance in respiratory disease management in Uganda: a qualitative study of patient and healthcare professional perspectives

Keir EJ Philip [1,2] Lucy L Cartwright,[3] Debra Westlake,[3] Grace Nyakoojo,[4] Ivan Kimuli,[4] Bruce Kirenga,[4] Evelyn A Brakema,[5] Mark W Orme [6] Daisy Fancourt [7] Nicholas S Hopkinson,[1] Rupert Jones,[8] Winceslaus Katagira [4]

For numbered affiliations see end of article.

**Correspondence to**
Dr Keir EJ Philip;
k.philip@imperial.ac.uk

## ABSTRACT

**Introduction** Music and dance are increasingly used as adjunctive arts-in-health interventions in high-income settings, with a growing body of research suggesting biopsychosocial benefits. Such low-cost, low-resource interventions may have application in low-resource settings such as Uganda. However, research on perceptions of patients and healthcare professionals regarding such approaches is lacking.

**Methods** We delivered sample sessions of music and dance for chronic respiratory disease (CRD) to patients and healthcare professionals. Seven participants took part in one singing and dance sample session. One patient completed only the dance session. We then conducted an exploratory qualitative study using thematic analysis of semistructured interviews with healthcare professionals and patients regarding (1) the role of music and dance in Ugandan life and (2) the perceived acceptability and feasibility of using music and dance in CRD management in Uganda.

**Results** We interviewed 19 participants, made up of 11 patients with long-term respiratory conditions and 8 healthcare professionals, who were selected by purposeful convenience sampling. Four key themes were identified from interview analysis: music and dance (1) were central components of daily life; (2) had an established role supporting health and well-being; and (3) had strong therapeutic potential in respiratory disease management. The fourth theme was (4) the importance of modulating demographic considerations of culture, religion and age.

**Conclusion** Music and dance are central to life in Uganda, with established roles supporting health and well-being. These roles could be built on in the development of music and dance interventions as adjuncts to established components of CRD disease management like pulmonary rehabilitation. Through consideration of key contextual factors and codevelopment and adaptation of interventions, such approaches are likely to be well received.

### Strengths and limitations of this study

► This is the first study to explore the perspectives of patients and healthcare professionals regarding the use of music and dance in respiratory disease management in Uganda.
► Using indepth interviews, triangulated with structured observations and key documentation, enabled a detailed, highly contextualised exploration of themes.
► Purposeful convenience sampling ensured appropriate representation from relevant stakeholders.
► As a single-site study, the transferability of findings cannot be ascertained.
► The COVID-19 pandemic and related restrictions regarding group activities, including singing and exercise, may have influenced the perspectives on participating in singing and dance activities.

asthma and chronic obstructive pulmonary disease (COPD) are leading causes of morbidity and mortality globally.[1] The burden of respiratory disease disproportionately affects people in low-income and middle-income countries, where over 90% of global respiratory deaths occur,[1–3] and is predominantly caused by smoking, respiratory infections, biomass smoke exposure, poor nutrition and air pollution.[2] Prevalence data are limited from Africa[4]; however, specifically in Uganda, research suggests that CRD are common.[5 6] People with CRD in Africa suffer from a high burden of symptoms amplified by social isolation, economic disadvantage and stigmatisation[7] related to their symptoms. Physical exercise training and self-management education are important components of CRD management, but can be challenging due to the interaction between symptoms, inactivity and

## INTRODUCTION

Chronic respiratory diseases (CRD) such as post-tuberculosis lung disease (PTBLD),

psychological impairment, which are key factors in the 'cycle of decline'.[7] There is interest in developing locally adapted, low-cost, high-impact interventions in this patient group. For example, a recent programme development study has shown that pulmonary rehabilitation (PR) is feasible and improves quality of life and exercise capacity in people with PTBLD in Uganda.[8]

Singing and dance have become increasingly popular adjuncts to conventional disease management strategies for people with long-term respiratory conditions in the UK[9] and other high-income countries.[10] Existing research suggests participants experience a range of biopsychosocial benefits, including those related to physical performance, mental health and well-being, and social isolation.[9 11–14] Music as distractive auditory stimuli during exercise training for people with CRD can reduce breathlessness and increase exercise capacity.[15] Although a large and growing body of research supports using arts to support health and well-being,[16] research in low-resource settings is limited. Additionally, coproduction of arts-in-health activities is widely appreciated as central to the successful development and adaptation of interventions.[17] Furthermore, a recent systematic review and meta-analysis of critical factors required for successful implementation of lung health interventions in low-income and middle-income countries emphasised the importance of ensuring compatibility with the local context and understanding the needs of local users.[18] Through the engagement of key stakeholders, including staff, patients and family members, such activities have the potential to use pre-existing sociocultural resources and minimise dependence on additional external funding or resources.

Data from this study examined the perspective of people with long-term respiratory conditions and respiratory healthcare professionals in Uganda to answer the following questions:

▶ What are the current roles of music and dance in general life?
▶ Would the use of music and/or dance in the management of long-term respiratory conditions be acceptable and feasible?

Answering these questions is important to establish if such approaches could be appropriate in Uganda, and if so inform the codevelopment of arts-in-health interventions.

## METHODS
### Research design
We conducted an exploratory qualitative study using thematic analysis. Data were collected using semistructured interviews with healthcare professionals and patients which focused on two main topics: (1) the role of music and dance in Uganda; and (2) the potential use of music and dance in CRD management in Uganda.

### Setting
The study was conducted at the Makerere University Lung Institute (MLI) outpatient clinic in central Kampala, Uganda. This urban setting was selected due to the trusted relationships between the patients, clinical staff and research teams, and the well-established academic relationship between the various research groups involved. Additionally, the institute has a well-established PR programme with informant groups that are knowledgeable regarding our topics of interest.

### Participants
Purposeful convenience sampling was used to ensure a representative sample of relevant individuals and groups by gender, age and religion, for both patients and healthcare professionals. Potential study participants were approached verbally (face-to-face or over the phone) after being identified by local staff and research team members working at the study location. Snowball sampling was used for further participant identification. Potential participants were provided with information (in appropriate language and format) about the study and then given time to consider if they wanted to participate.

#### Inclusion criteria
▶ Adults aged ≥18 years with CRD who attend, previously attended or have been invited to attend PR.
▶ Health professionals who work with people with CRD.
▶ Family members of adults with CRD.

#### Exclusion criteria
▶ People unable to give informed consent.
▶ People unable to participate due to physical or mental disabilities.

### Sample singing and dance sessions
Sample sessions took place in the same week as the interviews to give participants an idea of how the sessions could be structured and provide them participation experience. Trial singing sessions were delivered by Francis Mutesasira, a professional singing teacher. Francis is trained in the Singing for Lung Health methodology[19–21] and developed and ran the project 'Singing for Breathing (SFB) Uganda' for 3 months in 2018[22] at MLI Kupumua House, which consisted of Singing for Lung Health (Singing for Lung Health | British Lung Foundation; blf.org.uk) techniques adapted to local songs and vocal exercises. Sessions included relaxation and physical awareness exercises, physical warm-up, breathing exercises, song repertoire selected collaboratively with participants, and warm-down relaxation. Dance sessions were led by the lead physiotherapist for PR at the MLI, who regularly integrates dance movements into his rehabilitation sessions, and KEJP, who has developed and run dance sessions for people with long-term conditions. Sessions included a warm-up using simple rhythmic stepping, progressively demanding dance movements, selected and created collaboratively with participants, followed by a warm-down and gentle stretching. The intensity of the sessions

was continually adjusted to participants' perceived exertion levels. Sample sessions lasted between 20 and 40 min and took place at the MLI in a large room normally used for the exercise component of PR sessions.

## Data collection

Semistructured interviews were conducted in October 2019 at the MLI, in private rooms, with no non-participants present. The topic guide was developed by reviewing conceptually related research projects conducted by the team and others (see online supplemental file 1). Interviews focused on open-ended questions, with participant prompts to encourage further discussion on topics which appeared meaningful. Interviews were conducted by KEJP, LLC and GN, in English or Luganda (predominant local language) depending on the participant's preference. If in Luganda, GN, an experienced qualitative researcher, translated simultaneously. Interviews were audio-recorded, and interviewers documented immediate reflections following interviews. Interviewer–participant relationships were established through relaxed introduction, and participants were informed the interviewers were health professionals, but not directly involved in the provision of their individual healthcare. Modified Medical Research Council (mMRC) breathlessness scores were self-rated by patient participants using the mMRC scale, with options read out loud by the interviewer.

Structured observations of trial singing and dance sessions were conducted by KEJP, LLC and GN (see online supplemental file 2), and relevant documents were analysed (online supplemental file 3) to support contextualisation and interpretation of interview data.

Daily meetings took place involving (depending on availability) GN, IK, KEJP, LLC, RJ, BK and WK (DW from the UK), during which ongoing data collection and interpretation were discussed and triangulated with interview notes, structured observations and preparatory reference materials. This process aimed to facilitate understanding and inform the iterative development of ongoing data collection activities.

The participants were informed of the intention and focus of the research, and that their responses in no way influenced their ongoing care, rather that the intention was to inform the development of future interventions, if appropriate. Data were collected and handled as per the guidelines of Consolidated criteria for Reporting Qualitative research.[23]

## Data analysis

Interviews were transcribed verbatim. KEJP, LC and DW conducted a thematic analysis based on that described by Clarke *et al*.[24] and Terry *et al*.[25] During phase 1, transcripts were read and reread, with further listening and familiarising with interview recordings, interviewer reflections and structured observations. Importantly, notes from discussions between GN, IK, KEJP, LLC, RJ, BK and WK made during data collection were used to facilitate understanding. Phase 2 included open free-coding, discussion,

double-coding, cross-case analysis and development of coding structure. As such the analysis was predominantly inductive in nature, although deductive elements were contributed by the semistructured nature of using a topic guide. The coding structure was then refined into preliminary themes (phase 3), which were further discussed, refined, named and agreed on (phases 4 and 5). Participant validation was performed with staff members at the MLI. Given current COVID-19 restrictions, further patient participant validation was not performed; however, clarity and interparticipant consistency of identified themes suggest that further participant validation would have been unlikely to dramatically alter findings. Theme saturation was achieved during the analysis; however, given the exploratory nature of the study, all data were analysed. Coding and theme development used Microsoft Excel. Demographic and disease-specific information was sought from patient participants. Breathlessness scores were completed as an indication of disease-related functional impairment. This was selected given the heterogeneity of lung conditions represented; hence, a generic rather than disease-specific assessment was appropriate. Additionally, breathlessness is a key assessment criterion for PR and hence relevant for the application of this study's findings.

## Patient and public involvement

Participant feedback collected during the 'Singing for Breathing Uganda' project evaluation, combined with consultation with patients attending respiratory clinics, prompted this study and informed the topic guide development. Additionally, the primary objective of the study is an exploration of patient and healthcare provider perspectives and hence patient and public involvement is at the core of this study.

## Consent

All participants provided written informed consent.

## RESULTS

Nineteen participants were included in the study, made up of 11 patients and 8 staff members who were approached and recruited. Two further patients were approached and declined, stating they did not have time. Of the patient participants, 8 of the 11 were female, with a mean age of 43 years (range 20–63). Regarding ethnicity, all participants were black Ugandan. All patient participants reported CRD, including PTBLD (×6), post 'infection' lung disease (×1), asthma (×2), COPD (×1) and pulmonary fibrosis (×1). mMRC scores ranged from 1 to 3 (mean 1.5). None of the patients used ambulatory oxygen. Various symptoms were reported by patient participants, in keeping with their CRD, including breathlessness, cough and physical activity limitations. All reported living in houses (rather than flats or 'other'). Two lived alone and nine were cohabiting with family. Seven were in paid employment, one was a student and three were unemployed. In

order to gain the perspectives of individuals with varying amounts of exposure to these kinds of interventions, two patients were interviewed prior to sample sessions, while the other nine were interviewed after attending sample sessions, and two of these had also attended SFB Uganda the previous year.

Of the eight healthcare professionals, four were women and four were men, with a mean age of 41 years (range 29–59). Occupations represented were physiotherapist, respiratory researcher, administrator, carer (sister of a patient), nurse and three were doctors. Participant quotes are followed by 'P' for patient or 'S' for staff and the participant number.

On most topics, perspectives between patients and healthcare professionals aligned closely. Our analysis identified four key themes: music and dance as (1) central components of daily life; (2) having an established role supporting health and well-being; and (3) perceived as having strong therapeutic potential in respiratory conditions. However, the potential realisation of this 'strong potential' (theme 3) was dependent on theme (4): modulating demographic considerations of culture, religion and age.

### Theme 1: music and dance as central components of daily life

Music and dance were described as omnipresent in the social, religious and cultural components of daily life in Uganda. Music and dance were largely inseparable from one another and described as inclusive and participatory:

> Music really is everywhere for us…Music is really part of our fabric as a society….when they play a song everyone identifies to and everyone is getting up and just dancing, it doesn't matter whether they're in a suit, they're jumping, dancing. (S1)

> My wife is a politician, when we go to rallies, they normally invite you to come and join them. We join them. Yes. We join them and dance. (P4)

Music's omnipresence was attributed to its multiple social functions, especially forging interpersonal connectivity:

> Dancing is a way of communing, of interacting with people. It is one of those things that bind people. (S1)

> There is that kind of relationship, with people you sing with. (P3)

> Music speaks to our situations or just that feeling of being together with people and you're singing and you're dancing. (S3)

A further function being information transfer:

> Music is very important in our society because it gives messages, it educates through music you are able to know what is good, what is bad, what can be done, what happened in the past, what will happen in the future, all can be delivered through music. (P11)

Participation in music and dance was generally referred to as a free-willed choice; however, many also described compulsion, as if driven by an external 'power of music' that overcomes inhibiting factors:

> [I] feel the music in [me] and [have] to dance. (P9)

### Theme 2: music and dance had an established role supporting health and well-being

Through their role in social, religious and cultural aspects of life, music and dance were seen as already having established roles in supporting physical, mental and social health. Such effects were often described as concurrent and interrelated.

#### Mental health

The most dominant established health-promoting roles related to mental health. Most patients identified this function:

> [listening to music] you feel happy, you feel you are getting connected with the world that you are not seeing. It gives you some hopeful times. It gives a message. I keep with the message that gives some hope for the future. (P3)

> Instead of getting angry, [I] would try and find comfort in singing and dancing to control [my] anger. (P9)

Healthcare professionals also highlighted these functions for patients, but also frequently described using music for stress relief and relaxation themselves. Psychological benefits were underpinned by enjoyment of participation:

> I feel nice when I'm singing. (P8)

> When we are singing, of course you feel like you… you feel that joy. (S8)

#### Physical health

Physical health improvements were mainly attributed to dancing or exercising to music:

> Now [dance] has been taken up as one of the things that's used for physical exercises. (S4)

A group of doctors had also started an afterwork exercise group where they use music for working out, with dance often seen as preferential to other forms of physical activity.

> I don't like walking, if I have a car, I will drive it. Even to the nearest distance. But I would do dancing as a physical activity and I would do it with love. Because I love it and I love music. (S4)

Compared with dance, purely physical health benefits were not frequently attributed to singing in its established (daily life) roles; however, potential physical benefits of singing were mentioned in relation to singing used in a therapeutic context with patients (see theme 3).

## Social benefits

Social connectivity, as described in theme 1, supported social health and overlapped with mental health and well-being.

> You are joining other people. You know, when you are a people orientated person, when you find people that are happy, you also become happy. (P4)

This was unpinned by the light-hearted enjoyable nature of music and dance participation.

> It's a fun activity. It's a fun bonding activity for us. Everyone dances whatever they have, silly strokes, and you're just laughing and having a good time. (S1)

## Theme 3: music and dance perceived as having strong therapeutic potential in respiratory disease management

### Contextually appropriate

Perceived potential for successful integration was clear, largely due to the ubiquity (theme 1) and established roles (theme 2) of music and dance in promoting health and well-being.

> Because of what our culture is we love partying, we love music, we love dancing, so I think if someone who is told that if you dance, if you sing it is going to improve your health I believe they will have no problem taking part of it. (S3)

Again, fundamental to the perceived potential was enjoyment and group participation:

> I think it's good to do it as a group. Because you encourage each other. I think it's also more fun, yeah, and then it makes it, you know, something which you've got faster, you move on longer. (P1)

Potential psychosocial impact on patients' health conditions was highlighted:

> No amount of medicine can give you that human connection, which is a very important part of management. (S1)

Potential therapeutic mechanisms for physical improvements were also suggested by healthcare professionals:

> You go beyond your tidal volume, in terms of reaching out your respiratory effort…if they keep doing this song then every other time they have some incremental effort required of their respiratory muscles. (S7)

> I feel it helps because it requires breath control, breathing in, breathing out and at the end it is fun… And of course they are learning also how to sing, how to control their breath, which in their own way helps their healing process and of course coping with the environment. (S8)

The potential for delivery with minimal resource requirements was emphasised as an important factor, particularly where resources were most limited.

## Health benefits

Comments regarding potential health benefits for patients with CRD related closely to the established roles of music and dance in wider society.

Physical benefits related to potential exercise training effects, which were seen as very important for people with CRD.

> With the singing, you feel the lungs, you know, get opened, you feel you breathe very well. You feel the body also, the body moves with the singing, and also dancing. It becomes more free. (P3)

Some participants in the sample sessions reported improvements in symptoms, although it is important to highlight that these are subjective reports and no objective assessment of impact took place.

> The sputum can come out very easily. (P3)

> That their breathlessness has reduced so they can work a bit longer than they used to. Most of them, that's what they are saying. (S5)

Improvements in physical symptoms were intimately linked to psychological impact:

> I was feeling a bit happier because I feel like I could breathe a bit better. (P2)

The role of social aspects within the sample sessions were noted as creating peer support:

> It gives them courage and also helps them for the rehabilitation that they're supposed to do. Friends encourage each other to exercise. So it ends up being very, very efficient for them. (S6)

### Enjoyment

Sample session participants were very positive about the experience, which was also noted by staff:

> People were excited, and they say that let us do this whenever we come. They have been so touched. At first we thought, what is this now? But at the end, it has been perfect. (P6)

As in theme 1, enjoyment was a facilitator of health impact and the novelty of the approach was noted positively. Additionally, as in theme 2, participants emphasised the need to adapt sessions to the specific participants of a session (see theme 4).

### Already happening

The lead physiotherapist for PR was already integrating dance into his sessions and reporting very positive responses:

> When you bring in a warm-up that is full of dancing and rhythmical, we see they are happy. (S2)

Also, one patient reported using music for disease specific self-management:

When I get attacks, I go in my room, and what do I put on? The radio. So, what am I doing? Listening to music. (P4)

### Theme 4: modulating demographic considerations

Participants emphasised that, for successful implementation, activities or interventions would need to be adapted to the specific participants of any one group and the group itself. Key factors for consideration to ensure appropriate content included culture and religion, age, gender, and extent of urbanisation. These factors were important for two reasons: first to ensure that no member of the group felt uncomfortable or excluded; and second, responsive contextualisation was seen as a tool to optimise engagement and enjoyment—by selecting songs, music or dances that had cultural or historical significance to the group, a sense of collective identity could be established. This would facilitate interpersonal interactions based on shared experience and knowledge. Such an approach was almost presented as being obvious by participants, as this was how music and dance are used in Uganda more broadly. Contextualisation and personalisation were seen as being part of the essence of music and dance themselves.

Dancing has no formula, it has no pattern. It's not a matter of, oh you must conform. Each one has their own dance. I believe that if I was dancing with you, you have your own style of dancing, and I have my own style of dancing. (S2)

### Culture and religion

For the study respondents, the concepts of culture and religion were interrelated. The terms 'culture' or 'traditional' were often used in reference to traditional tribal practices, beliefs and identities, while 'religion' referred to world religions (Christian, Muslim or atheist/agnostic).

Those folk songs, traditional, that people can engage to traditional dances that train from tribe to tribe. (S2)

Culture is extremely important in Uganda, and music and dance are core to these aspects of daily life (theme 1):

There is no culture in Uganda where there isn't dancing. (P1)

However, expressions and norms differ:

Every culture, every part of this country has a different kind of dance. (S1)

Similarly, religion is very important. In Kampala the majority of people identify as Christian, of various denominations, with a smaller but significant proportion following Islam (14%).[26] Music and dance are prominent in religious practices and contexts.

We rarely go direct into praising, praying without singing, without dancing… of course giving glory to god, giving your leg, you are giving your arms, so why not dance. (S8)

For the Christians, they are used to singing, because in churches, Protestants do sing. Catholics do sing. Adventists do sing. Born again, most of the people… even the Witch crafts they have their praise, they praise. Yeah. People are used to singing. And Muslims sometimes they do sing. (P6)

It was suggested that Muslim participants might find singing and dance less acceptable; however, one Muslim participant was positive about the sample session:

The dancing helps [me] so much, it's so uplifting. (P9)

Cultural norms were also highlighted, such as issues around exposing parts of the body in close proximity or how social status may influence acceptability and participation:

"I'm a Sheikh. I'm a Bishop. I'm a very tough father at home." You know, that kind of person who has a very cut-out social role they probably won't come to sing so much… Such a patient might think that singing might be lowering their social role. (S7)

Differences between urban and rural norms were highlighted:

In the rural areas dancing is more associated to ceremony party, not a day to day. (P1)

Rural areas are more conservative 'dresses that are longer, skirts like longer, no slits'. (P1)

### Age

As per theme 1, music and dance were described as having multiple functions; the predominant function for an individual was seen to be modulated by age.

The old people they still love their music. Where it's a story telling song or it's something to harmonise and move or to advance excitement at a party. Yeah generally the young people of course they love it. Dancing and shaking around. (S7)

However, age was not seen as a barrier:

[older people] like dancing, and quite many of the old they get excited and dance. (S7)

### Perception of others

The importance of these demographic factors also related to how participation might be seen by others, including family, friends and the wider community. Overall, if the activities were clearly being delivered in a therapeutic capacity, participants felt that social acceptance would be high.

[my family members] are excited, they want the results afterwards. (P8)

### Improving acceptability

Given these considerations, participants suggested various ways to optimise acceptability. An emphasis on dance being physical exercise was proposed. Additionally, clearly stating the intended therapeutic benefits was important. Similarly, the therapeutic intention of singing was important, and this was well communicated during the sample sessions:

The singing, it is a different kind of singing also, yes, not all songs. But just get songs that push the lungs, expands the lungs, makes the lungs okay, yeah. And the dancing, it depends on the strokes you make, there are dance strokes that stretches the muscles. (P8)

Health professionals felt acceptance would be more forthcoming if a clear evidence base was also provided, and using the local languages was described by one participant as a method of increasing engagement through cultural identification:

They will be more interested in the songs which are done in the local languages. They are richer in terms of connection with the audience. (S7)

Appropriate song selection would be facilitated by using secular music and cocreating session content specific to the group. This approach worked well in the sample sessions:

We sang our national anthem of Uganda. It is for all of us. (P6)

Of note, although the demographic variables highlighted were considered important by study participants, they were eclectic in their music preferences, with culture and religious norms seen as informing, rather than limiting:

The trend is from cultural, traditional, to any pattern somebody wishes to. (S2)

### Additional implementation factors

In addition to demographic considerations, there was a broad appreciation that session content would also be adapted to the physical capacity of individual participants. Participants in the sample sessions felt such adaption took place successfully:

For those who are a bit weak, to know that they can rest, when the body feels that it is tired. I thought that that was good. (P1)

Also, financial and time costs would need consideration to facilitate attendance. Suggestions included having sessions a maximum of once weekly and subsidising travel costs to ensure sessions led to net benefit rather than risk contributing to already strained financial situations.

## DISCUSSION

The results of this indepth qualitative study show that music and dance are core components of daily life in Uganda. Study participants felt that participation supports both collective and individual health and well-being. These functions supported the perspective, from patients and healthcare professionals, that music and dance had great potential to improve elements of physical, mental and social health and well-being for people with CRD. Individuals who had prior experience of arts-in-health activities or who participated in sample sessions were very enthusiastic about the concept. Those without prior experience could see value in the concept and were happy to try. They highlighted important factors for consideration for codevelopment and successful implementation primarily related to culture, religion and age.

This study has multiple strengths. First, to our knowledge it is the first to explore this topic. Second, the wide-ranging expertise of the research team strengthened interpretation. Third, using indepth interviews, triangulated with structured observations and key documentation, enabled a detailed, highly contextualised exploration of themes. Fourth, purposeful convenience sampling ensured appropriate representation from relevant stakeholders.

Certain study limitations and considerations are important to discuss. First, being a single-site study, with a sample of 19 participants, the transferability of findings cannot be ascertained, particularly regarding areas of Uganda outside of Kampala, where social and cultural groups and norms are likely to differ. However, Kampala is a district that has a mixture of all tribes in Uganda, and the MLI is a specialist centre receiving referrals from all over the country. Second, COVID-19 pandemic restrictions relating to certain group activities, including singing and exercise,[27] are currently in place, and these data were collected prior to the pandemic; hence, COVID-19-related concerns may change the experience of group activities such as music and dance when they are considered safe to recommence. Additionally, here we report participants' perceptions regarding potential health benefits, and although in general beneficial effects of similar interventions have been demonstrated, formal research and evaluation of this specific intervention are still required.

Although no other studies have investigated this topic in low-resource settings, our findings echo those of research in related contexts. Research on Singing for Lung Health in the UK suggests participants perceive a range of physical, psychological and social benefits in keeping with our findings.[11 14 19 21 28 29] Similarly, studies regarding the perceived impact of dance for people with long-term respiratory conditions in the UK and Canada identify a range of biopsychosocial benefits.[10 12 30] Additionally, an evaluation of SFB Uganda, a singing project for people with CRD in Uganda, provided anecdotal reports that participation was enjoyed,[22] and our findings are broadly in keeping with the evaluation of SFB. Similarly, anecdotal

experience of related singing and dance projects for people with long-term respiratory conditions in other low-resource settings showed participants enjoyed the experience and reported a range of biopsychosocial benefits.[31] A study of culturally adapted PR at the MLI also showed high levels of acceptability.[8] Importantly, in each situation described, contextual adaptation and codevelopment of activities appear crucial to success. Interestingly, there was close alignment regarding responses from patients and healthcare professionals. This may be expected with regard to the general role of music and dance in Uganda, but was also the case in relation to potential therapeutic interventions. The main differences between the groups were healthcare professionals discussing potential therapeutic mechanisms in more depth and emphasising the requirement for an evidence base to increase acceptance. Future research should include assessing the impact of participation on relevant health outcomes and physiological parameters, building on related physiological research already completed.[13]

The current COVID-19 pandemic has necessitated certain restrictions on group activities such as singing[27] and dancing, which are likely to impact the potential application of these findings in the short term. However, developments including widespread immunisations, infection control measures and remotely delivered singing and dance interventions[11 32 33] may help reduce risk. Additionally, although the majority of participants were highly positive about participation, there were exceptions. As such, music and dance could be used as optional adjuncts to optimise uptake and completion of established, evidence-based respiratory management approaches such as PR.

## CONCLUSIONS

Long-term respiratory conditions are common in Uganda, causing a high burden of morbidity and mortality. Low-cost, low-resource interventions are of wide-reaching interest. Our findings suggest people with CRD and healthcare professionals see a great deal of potential in the use of music and dance as adjunctive roles to PR, or possibly be delivered as independent activities within CRD management. Building on established therapeutic roles of music and dance in wider Ugandan society, through coproduced intervention development specific to respiratory patients, appears to be a viable route for intervention development. These findings are important for developing arts-in-health interventions in Uganda and beyond.

**Author affiliations**
[1]National Heart and Lung Institute, Imperial College London, London, UK
[2]NIHR Imperial Biomedical Research Centre, Imperial College London, London, UK
[3]Faculty of Health, University of Plymouth, Plymouth, UK
[4]Makerere University Lung Institute, Makerere University, Kampala, Uganda
[5]Department of Public Health and Primary Care, Leiden University Medical Center, Leiden, The Netherlands
[6]Department of Respiratory Sciences, University of Leicester, Leicester, UK
[7]Department of Behavioural Science and Health, University College London, London, UK
[8]PUPSMD, University of Plymouth, Plymouth, UK

**Acknowledgements** We would like to thank the study participants for their time and effort, Stuart Blatston at the 625 for supporting logistical requirements of the project, Francis Mutesasira for delivering the singing sessions and Richard Kasiita for his contributions.

**Contributors** KEJP, RJ, WK and BK had the original idea for the study. All authors were involved in designing the study. DW provided guidance on qualitative methods, including design and analysis. KEJP, LLC and GN conducted the interviews. KEJP, LLC, DW, GN, RJ and WK conducted the analysis, which was then discussed with the other authors and refined. The first draft of the manuscript was written by KEJP. All authors read, contributed to and agreed on the final manuscript draft.

**Funding** KEJP was supported by the National Institute for Health Research Academic Clinical Fellowship award and the Imperial College Clinician Investigator Scholarship. DF was supported by the Wellcome Trust (205407/Z/16/Z). The research leading to these results has received support from the EU Research and Innovation Horizon 2020 programme (H2020 European Research Council) under grant agreement number 680997, part of the Health, Medical Research and the Challenge of Ageing Horizon 2020 action, and funding from the University of Plymouth Global Challenges Research Fund (GCRF) internal fund.

**Disclaimer** The funders had no say in the design and conduct of the study; collection, management, analysis and interpretation of data; preparation, review or approval of the manuscript; and decision to submit the manuscript for publication. This publication presents independent research. The views expressed are those of the authors and not necessarily those of the NHS, the NIHR or the Department of Health and Social Care.

**Competing interests** RJ declares grants unrelated to this study from AstraZeneca, GlaxoSmithKline and Novartis, and personal fees for consultancy, speakers fees or travel support from AstraZeneca, Boehringer Ingelheim, GlaxoSmithKline, Novartis, Nutricia and OPRI. No other authors declare conflicts.

**Patient consent for publication** Not required.

**Ethics approval** Ethical approval was granted by the Mulago Hospital Research and Ethics Committee (reference number MHREC: 1478) and the University of Plymouth Faculty Research Ethics and Integrity Committee (19/20-1164). All research activities were conducted in accordance with the principles of the Declaration of Helsinki.

**Provenance and peer review** Not commissioned; externally peer reviewed.

**Data availability statement** All data relevant to the study are included in the article or uploaded as supplementary information.

**ORCID iDs**
Keir EJ Philip http://orcid.org/0000-0001-9614-3580
Mark W Orme http://orcid.org/0000-0003-4678-6574
Daisy Fancourt http://orcid.org/0000-0002-6952-334X
Winceslaus Katagira http://orcid.org/0000-0003-4622-191X

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
