## [Reviewer comments · BMJ Open]

ARTICLE DETAILS

TITLE (PROVISIONAL)	Music and Dance in respiratory disease management in Uganda: A qualitative study of patient and healthcare professional perspectives
AUTHORS	Philip, Keir; Cartwright, Lucy L; Westlake, Debra; Nyakoojo, Grace; Kimuli, Ivan; Kirenga, Bruce; Brakema, Evelyn A; Orme, Mark; Fancourt, Daisy; Hopkinson, Nicholas; Jones, Rupert; Katagira, Wincelous

VERSION 1 – REVIEW

REVIEWER	Giuseppe Sanna Sassari University Hospital
REVIEW RETURNED	30-May-2021

GENERAL COMMENTS	In this paper Keir et al. investigated the role of music and dance in respiratory disease management in patients in a low-income Country. Although I understand that this paper should be considered preliminary, in my opinion it has some limitations: 1) The sample size (the Authors talk about a convenience sampling) is really small (11 patients).2) The beneficial effects of music listening are well known in several medical conditions (e.g. heart failure) and they are related to: heart rate, blood pressure and respiratory modulation, modulation of neuroendocrine system, behavioral effects. Thus, the finding of "positive" effects in patients in a Country where music and traditional dance are a fundamental part of the cultural behavior is not unexpected.3) I understand that Authors performed a qualitative study; however, the reader is expected to find some "consistent" data about the effects of music and dance on respiratory parameters in the enrolled patients. Moreover, therapies and respiratory capacity of the enrolled patients are missing (the Authors only say that none of the patients was on oxygen therapy).
--

REVIEWER	Rachel Goldenberg Western University
REVIEW RETURNED	16-Jun-2021

GENERAL COMMENTS	This is a well-designed study that makes an excellent and valuable contribution to the literature. The content is well presented and the topic will be interesting to many. My revisions are relatively minor suggestions and I wish to congratulate the authors. 1. Could you use a different term from "taster sessions?" I find this too colloquial. Perhaps sample sessions or trial, as you use later on. Regardless, be consistent and a bit more formal. Could you also explain the content of the sessions more? I am curious about the
--

	singing and dance methods. Were they separate sessions or integrated? How many were there? If they were separate, did everyone attend both? Please flesh out this section and clarify. 2. In Data Analysis: How did you determine breathlessness scores? I see it's mentioned in the results, but could you put it here too? 3. In Results: Please clarify the sentence "Eight of the eleven participants were female..." Do you mean patient participants? Could you separate the participant demographics from patients versus staff? Or at least make it clearer and easier to distinguish. Also comment on the two patients interviewed prior to the class. Why was this? Could you add something in the discussion? Finally, it might be worth mentioning prior formal experience with PR or singing and dance. 4. Under Theme 2, physical health: You say relative to dance, people didn't comment on singing. Did anyone mention physical benefits of singing? What did they say? Perhaps reword this. 5. Theme 3, health benefits: It will be important to report how many sessions were attended, especially when reporting outcomes like sputum movement and reduced breathlessness. 6. Theme 4: Omit or change the sentence "It was suggested that Muslim participants might find singing..." Find a way to report this finding more generally. I think it's enough to say something like "the singing and dance was acceptable in all cultural contexts that included Christians and Muslims..." I am unclear about P1 rural areas. Perhaps this is a typo? 7. Discussion: Regarding the statements about COVID-19 in the fifth paragraph "The current Covid-19 pandemic has necessitated...", I feel this is too speculative. It's not irrelevant, but might not be necessary. I think it is worth pointing out that these data were collected prior to covid, as you do earlier. You mention there were exceptions to the positive perceptions. Can you report these? Thank you for your work!
--	--

VERSION 1 – AUTHOR RESPONSE

Reviewer: 1

In this paper Keir et al. investigated the role of music and dance in respiratory disease management in patients in a low-income Country.

Although I understand that this paper should be considered preliminary, in my opinion it has some limitations:

- 1) The sample size (the Authors talk about a convenience sampling) is really small (11 patients).

Many thanks for raising this point. We realise that this was not as clear as it could have been, our study explored the perspectives of both patients (n=11) and healthcare professionals (n=8), providing a sample size of 19. Our use of in-depth individual semi structured interviews with 19 participants provided a large amount of rich and complex qualitative data for interpretation. Our sampling strategy proved this to be adequate to achieve theme saturation. As highlighted, we used convenience

sampling, which was appropriate for the research topic, and by using a purposeful approach, we were able to be confident that the relevant stakeholders were included.

However, we realise that this should be clarified in the manuscript and acknowledged when considering the extent to which our findings might be applied in other contexts. As such, we have clarified the total participant number in the abstract, 1st line of the results, and edited the first sentence of the limitations and considerations section in the discussion (3rd paragraph): 'Firstly, being a single site study, with a sample of 19 participants, the transferability of findings cannot be ascertained....'.

2) The beneficial effects of music listening are well known in several medical conditions (e.g. heart failure) and they are related to: heart rate, blood pressure and respiratory modulation, modulation of neuroendocrine system, behavioural effects. Thus, the finding of "positive" effects in patients in a Country where music and traditional dance are a fundamental part of the cultural behaviour is not unexpected.

Our study did not aim to find the unexpected, rather to carefully and scientifically examine a topic of relevance to the development of novel, contextually and culturally appropriate, evidence-based interventions. We agree that there is a substantial body of research on the impacts of music listening in various conditions, but would like to emphasise that we did not intend to conduct a study of the effect of these activities, but instead, to explore the perspectives of the relevant stakeholders to inform the implementation of such approaches in this context. Our focus was driven by our prior experience of work in Uganda indicating the role of religion, culture and age in beliefs and perceptions about treatment options. Therefore, we felt that the direct application of research from other settings would not have been appropriate. Hence, developing a more nuanced understanding of the topic remained valuable, and was the intention of our study. Our view on this point is further supported by a recent systematic review and meta-analysis, by members of our group and others, assessing critical factors required for successful implementation of lung health interventions in low and middle-income countries, published in the European Respiratory Journal (doi: 10.1183/13993003.00127-2020), which states, among the key factors are ensuring compatibility with the local context and understanding needs of local users.

To clarify the importance of the research topic we have added the following to the second paragraph of the introduction 'Furthermore, a recent systematic review and meta-analysis of critical factors required for successful implementation of lung health interventions in low and middle-income countries emphasised the importance of ensuring compatibility with the local context and understanding needs of local users¹⁸.'

3) I understand that Authors performed a qualitative study; however, the reader is expected to find some "consistent" data about the effects of music and dance on respiratory parameters in the enrolled patients. Moreover, therapies and respiratory capacity of the enrolled patients are missing (the Authors only say that none of the patients was on oxygen therapy).

Thank you for this comment. We agree that research assessing effects of interventions and physiological responses to participation would also be valuable, so have highlighted this in the discussion as important considerations for future research, through the addition of the following to the manuscript discussion 'Future research should include assessing the impact of participation on relevant health outcomes and physiological parameters.'

We decided not to include these measurements in our study as it was our opinion that if we were to include pre and post physiological measures it would have fundamentally changed the experience of the participants, which in itself may have impacted the primary focus of the study – namely, their perspectives on the use of music and dance as components of their management. As such, to avoid negatively impacting the integrity of the results, we felt it best to focus on the topic with the qualitative approach used. That said, we very much agree such topics are important and have been working on various other projects to explore those questions, such as this one: [Physiological demands of singing for lung health compared with treadmill walking | BMJ Open Respiratory Research \(doi: 10.1136/bmjresp-2021-000959\)](https://doi.org/10.1136/bmjresp-2021-000959).

Regarding further description of the participants and their disease characteristics, we feel that the information included is appropriate for the focus of the research topic. The first paragraph of the results outlines the relevant information regarding factors that are likely to impact their experience and perspectives which was the focus of the study. Regarding 'respiratory capacity', we felt the mMRC breathlessness scale was more appropriate as this is more relevant to the participants experience and opinions than spirometry results. Similarly, it is unclear how listing a patient's medications or other therapies would have made a useful contribution to the interpretation of the interviews. We agree that data on medications and spirometric assessments can be useful, but this would be more relevant to a study assessing impacts of participation on health parameters, and as such, we felt it was less relevant here.

Reviewer: 2
Dr. Rachel Goldenberg, Western University

Comments to the Author:

This is a well-designed study that makes an excellent and valuable contribution to the literature. The content is well presented and the topic will be interesting to many. My revisions are relatively minor suggestions and I wish to congratulate the authors.

Many thanks.

1. Could you use a different term from "taster sessions?" I find this too colloquial. Perhaps sample sessions or trial, as you use later on. Regardless, be consistent and a bit more formal. Could you also explain the content of the sessions more? I am curious about the singing and dance methods. Were they separate sessions or integrated? How many were there? If they were separate, did everyone attend both? Please flesh out this section and clarify.

We have changed this to 'sample sessions' as suggested. We have also added to the description of the sample sessions in the methods section, and added a link to more information on the Singing for Lung Health approach.

2. In Data Analysis: How did you determine breathlessness scores? I see it's mentioned in the results, but could you put it here too?

Thank you for highlighting this, we have added the following to the data collection section 'Modified Medical Research Council (mMRC) breathlessness scores were self-rated by patient participants using the mMRC scale, with options read out loud by the interviewer.'

3. In Results: Please clarify the sentence "Eight of the eleven participants were female..." Do you mean patient participants? Could you separate the participant demographics from patients versus staff? Or at least make it clearer and easier to distinguish. Also comment on the two patients interviewed prior to the class. Why was this? Could you add something in the discussion? Finally, it might be worth mentioning prior formal experience with PR or singing and dance.

Thank you for highlighting this. We have separated and clarified the descriptions of the patient participants from the staff participants.

We interviewed two patients prior to the class to provide perspectives from people with differing levels of prior experience. This has been clarified in the manuscript with the addition of the following 'In order to gain the perspectives of individuals with varying amounts of exposure to these kind of interventions, two patients were interviewed prior to sample sessions, while the other nine were interviewed after attending sample sessions, and two of these had also attended SFB Uganda the previous year.'

We have also added a comment to the first paragraph of the discussion 'Those without prior experience could see value in the concept and were happy to try.'

4. Under Theme 2, physical health: You say relative to dance, people didn't comment on singing. Did anyone mention physical benefits of singing? What did they say? Perhaps reword this.

Comments regarding the physical impacts of singing in a general social (non-therapeutic) context were very limited, so we did not want to bring a focus to these comments that was not supported in the data. However, as suggested we have reworded this to the following 'Compared with dance, purely physical health benefits were not frequently attributed to singing in its established (daily life) roles, however potential physical benefits of singing were mentioned in relation to singing used in a therapeutic context with patients (see Theme 3).'

5. Theme 3, health benefits: It will be important to report how many sessions were attended, especially when reporting outcomes like sputum movement and reduced breathlessness.

We have clarified this point in the methods adding 'We delivered sample music and dance for chronic respiratory disease (CRD) sessions to patients and healthcare professionals. Seven participants took part in one singing and dance sample session, which included one patient completed only the dance session.'

We have also further emphasised the subjective nature of the comments reporting symptoms improvements related to sample session participation 'Some participants in the sample sessions reported improvements in symptoms, though it is important to highlight that these are subjective reports, and no objective assessment of impacts took place.'

6. Theme 4: Omit or change the sentence "It was suggested that Muslim participants might find singing..." Find a way to report this finding more generally. I think it's enough to say something like "the singing and dance was acceptable in all cultural contexts that included Christians and Muslims..."

We have removed this sentence, but kept the quote, which demonstrates the point.

I am unclear about P1 rural areas. Perhaps this is a typo?

Thank you for highlighting this. We have clarified the presentation of this quote from participant 1.

7. Discussion: Regarding the statements about COVID-19 in the fifth paragraph "The current Covid-19 pandemic has necessitated...", I feel this is too speculative. It's not irrelevant, but might not be necessary. I think it is worth pointing out that these data were collected prior to covid, as you do earlier.

Thank you for raising this. We have highlighted these comments are speculative with the addition of the following ' However, currently we can only speculate about how the COVID-19 pandemic will impact the perceptions of relevant stakeholders regarding participation in such activities in the future.', but feel it is important to highlight that such factors are likely to modulate the findings of our study moving forward, so are worthy of consideration.

You mention there were exceptions to the positive perceptions. Can you report these?

Here we were referring to the various considerations highlighted by participants regarding making the sessions appropriate for all participants. We have clarified this point as follows 'Additionally, although the majority of participants were highly positive about participation, multiple considerations were highlighted regarding acceptability, and as such it might not always be possible to adapt sessions for all participants, all the time.'

Thank you for your work!

Thank you for taking the time to review it!